# Investigating the incidence and risk factors of hypertension: A multicentre retrospective cohort study in Tabuk, Saudi Arabia

**Umar Yagoub**[1]*, **Nasrin S. Saiyed**[1], **Bandar Al Qahtani**[2], **Attiya Mohammed Al Zahrani**[3], **Yassir Birema**[4], **Ibrahim Al Hariri**[5]

**1** Research Department, King Salman Armed Forces Hospital, Tabuk, Saudi Arabia, **2** Academic Affairs Department, King Salman Armed Forces Hospital, Tabuk, Saudi Arabia, **3** Surgery Department, King Salman Armed Forces Hospital, Tabuk, Saudi Arabia, **4** Cardiology Department, King Salman Armed Forces Hospital, Tabuk, Saudi Arabia, **5** Family Medicine Department, King Salman Armed Forces Hospital, Tabuk, Saudi Arabia

* mohammedumar2001@yahoo.com

**Data Availability Statement:** The data used to support the findings of this study are available from King Salman Armed Forces Hospital and King

## Abstract

### Background

Hypertension is a major global health concern affecting approximately 1.13 billion people worldwide, with most of them residing in developing countries. The aim of this study was to determine the incidence of different stages of hypertension and its associated modifiable and non-modifiable risk factors among patients in military-setting hospitals in Tabuk, Saudi Arabia.

### Methods

This retrospective cohort study was conducted at two hospitals in Tabuk, Saudi Arabia. The data were collected from hospital electronic records from 1 January 2019 to 31 December 2019. The blood pressure levels of patients from the last three separate medical visits were recorded. Descriptive statistics and multinomial logistic regression were used for the data analysis.

### Results

The study included 884 hypertensive patients. The incidences of stage of elevated BP, stage 1, stage 2, and hypertension crisis were 60.0, 29.5, 7.0, and 3.5 cases per 1000 persons. Multivariate analysis indicated that progression from the stage of elevated blood pressure to hypertension crisis was significantly associated with advanced age (odds ratio [OR] = 3.62, 95% confidence interval [CI] = 1.99–8.42), male sex (OR = 2.84, 95% CI: 0.57–5.92), and a positive family history of hypertension (OR = 1.95, 95% CI: 1.23–3.09). Other key determinants of the development of stage of elevated blood pressure to hypertension crisis were current smoking status (OR = 1.74, 95% CI: 1.23–4.76), and physical inactivity (OR = 6.48, 95% CI: 2.46–9.14).

Khalid Military Hospital but restrictions apply to the availability of these data, which were used under license for the current study, and so are not publicly available. Data are however available from research and ethics committee at King Salman Armed Forces Hospital (contact via telephone number: +966 14 4411088 extensions:84571 and email:res_proposal@nwafh.med.sa) for researchers who meet the criteria for access to confidential data.

**Funding:** The study was fully funded by Research Unit, Academic Affairs Department, King Salman Armed Forces Hospital Tabuk, Kingdom of Saudi Arabia. Grand no. KSAFH/AFD/006, Dr Attiya Mohammed Al Zahrani.

**Competing interests:** NO authors have competing interests.

## Conclusion

The incidence stage of elevated blood pressure was high among the patients investigated at armed forces hospitals in Tabuk, Saudi Arabia. The logistic regression model proposed in the present study can be used to predict the development of different stages of hypertension. Age, sex, marital status, family history, smoking status, and physical activity play an important role in the development of hypertension. Better strategies to improve awareness, screening, treatment, and management of hypertension are required in Saudi Arabia.

## Introduction

Hypertension is a common noncommunicable disease with a massive global impact. It increases the risk of several medical conditions such as cardiac diseases, congestive cardiac failure, stroke, retinal haemorrhage, and kidney diseases [1]. Socioeconomic problems have changed the lifestyle of people in low- and middle-income countries, which has increased the risk of developing hypertension [2]. Rapid urbanisation in Saudi Arabia over the past two decades, has led to an increase in incidence of hypertension [3]. Research studies have shown that hypertension imposes direct economic burden on both the patient and the health care system, by increasing the need for hospitalisation, physician consultations, laboratory investigations, and prescribed medications [4].

Hypertension has a complex aetiology, with different incidence, prevalence, and fatality rates depending on the race and ethnicity. Several studies have investigated the incidence of hypertension with emphasis on different races and ethnicities; however, the results obtained were dependent on the definition of hypertension followed in the individual study. A prospective, population-based cohort study conducted in six communities in the United States reported that the crude incidence rate of hypertension per 1000 person-years, was 56.8 for whites, 84.9 for Blacks, 65.7 for Hispanics, and 52.2 for Chinese [5]. In Eastern Mediterranean Region countries, the incidence of hypertension has been revealed to reach up to 25% in the adult population, while in the American continent, this has been shown to range from 14% to 40% among people aged 35 to 64 years [6].

Several risk factors have been associated with hypertension. These factors vary from country to country, and even prone to variations in socio-demographics of the same region. Previous epidemiological studies have reported the determinants of hypertension as both modifiable and non-modifiable risk factors [7–9]. Studies in Saudi Arabian populations, have explored relevant risk factors for hypertension, including age, sex, sociodemographic factors, family history, and behavioural and lifestyle characteristics, but these studies were limited to cross-sectional data [10, 11]. Therefore, incidence of hypertension and associated risk factors need to be defined and addressed in a representative large sample of the rapidly expanding Saudi Arabian population.

King Salman Armed Hospital (KSAFH) and King Khalid Military Hospital (KKMH) are two military hospitals in the city of Tabuk in Saudi Arabia. An increase in the incidence of cardiovascular diseases, stroke, and renal diseases, such as chronic renal failure has been reported in these two hospitals in the last 5 years with hypertension being the leading cause of morbidity, mortality and a high number of hospital admissions in both the hospitals. This observation made it necessary to explore more about hypertension, its incidence, and associated risk factors. To the best of our knowledge, the current retrospective cohort study on hypertension and

related risk factors is the first study of this type to be conducted in this region and is vital for the establishment of successful hypertension prevention strategies. The results of this study can be extrapolated to other regions of Saudi Arabia as well as in the Gulf region which has similar demographics and an increased incidence of diseases which are considered to be complicated by hypertension.

## Materials and methods

### Study cohort

A hospital-based retrospective cohort study was conducted using a common registry program. Patients registered at KSAFH's nephrology, cardiology, and internal medicine departments and KKMH's family medicine clinic between 1 January 2019 and 31 December 2019 were selected. Both inpatients and outpatients from these departments during the study period were considered. The majority of hypertensive patients were referred to these departments for treatment of hypertension according to the hospital policy. All patients aged 18 years or older and diagnosed with hypertension according to the 2017 guidelines of American College of Cardiology/American Heart Association (ACC/AHA 2017) were included in this study cohort. A total of three blood pressure (BP) readings were taken for every patient during the study period. Patients with a diagnosis of hypertension for less than 6 months, pregnant women, and patients with missing medical information were excluded from this study. If a patient had more than three BP measurements, only the last three measurements were used. The study cohort consisted of exposed and non-exposed groups. Subjects with a mean BP measurement falling within a defined stage of hypertension were considered in the exposed group.

### Sample size

The sample size was calculated using the public domain software Epi Info version 7.2 developed by the Centers for Disease Control and Prevention. The parameters used in the sample size calculation were chosen arbitrarily owing to resource constraints. The parameters were two-sided significance level (1-alpha): 95, power (1-beta, the % chance of detecting): 80, ratio of sample size -unexposed/exposed: 1, percentage of unexposed with outcome: 5, percentage of exposed with outcome: 10, odds ratio: 2.1, risk/prevalence ratio: 2, and risk/prevalence difference: 5. Thereby, the calculated sample size was 950.

### Operational definitions

**Hypertension.** According to the 2017 guidelines on hypertension by the American College of Cardiology/American Heart Association (ACC/AHA 2017), hypertension was defined as systolic blood pressure (SBP) ≥130 mmHg, diastolic blood pressure (DBP) ≥80 mmHg or taking any prescribed antihypertensive drugs to control BP. For the present study, patients were classified into four stages according to their BP levels on the basis of the ACC/AHA guidelines [12]:

1. Stage of Elevated BP: SBP between 120–129 mmHg and DBP <80 mmHg;

2. Stage 1: SBP between 130-139 mmHg or DBP between 80-89 mmHg;

3. Stage 2: SBP ≥ 140 mmHg or DBP ≥ 90 mmHg;

4. Hypertension crisis: SBP > 180 mmHg and/or DBP > 120 mmHg.

**Risk factors for hypertension.** Hypertension does not have an exact cause except in case of secondary hypertension. Several factors affect the incidence of this disease. Some are modifiable, whereas others are not modifiable. Modifiable risk factors are mostly include lifestyle choices, such as smoking, poor physical activity, and employment status. However, some individuals are at a higher risk of hypertension, regardless of their lifestyle. These are non-modifiable risk factors, such as age, sex, marital status, race/ethnicity, and family history of hypertension [13–15]. In the current study, a family history of hypertension was defined as having one or more close family members with high BP and smoking status was defined as never smoked, past smoker, or current smoker. Physical activity was defined as exercise performance for at least 30 minutes per day for 5 days in a week.

**Incidence rate.** The incidence rate measures the frequency of new cases of disease in a population. It considers the sum of the time that each person remained under observation and at risk of developing the outcome under investigation [16]. It is calculated as:

$$Incidence\ rate = \frac{Number\ of\ new\ cases\ of\ a\ disease\ in\ a\ given\ time\ period}{Total\ person - time\ at\ risk\ during\ the\ follow - up\ period}$$

**Relative risk.** The relative risk or risk ratio is used when the study involves comparing the likelihood or chance of an event occurring between two groups [17]. It is calculated as:

$$Relative\ Risk = \frac{Probability\ of\ an\ event\ occuring\ for\ one\ group}{Probability\ of\ an\ event\ occuring\ for\ other\ group}$$

## Data collection tool and measurement

When examining patients, all physicians had access to the hospital's electronic records, which contained complete information of patient's previous visits, current medical conditions including their BP readings, and their current drug list and comorbidities. An appropriate diagnosis of hypertension was defined as ICD-10 code on the hospital electronic record system. A structured data collection tool was developed based on a comprehensive literature review, which consisted of baseline demographic information, medical history, and BP. All the questions in the tool were close-ended. The BP of the patients was recorded by well-trained nurses using an automatic BP monitor. The mean follow-up interval for the current study was 3 months.

## Data entry and data management

Data were collected from hospital electronic records by an experienced research assistant. The coding of continuous and categorical variables was performed and reviewed for any discrepancies. The data collection process was followed by two investigators, and the consistency of the entered data was cross-checked. Data that did not meet quality control inspections were excluded from the analysis. Data management and analysis were performed using the Statistical Package for Social Sciences version 23.0, for Windows (SPSS Inc., Chicago, USA).

## Statistical analysis

The results were summarised as frequencies and percentages for qualitative variables and means and standard deviations (SDs) for quantitative variables. The incidence rates and relative risk ratios for the different stages of hypertension were calculated. For analysis purposes,

the mean BP measurement of the three visits was used to assign the stages of hypertension. Multinomial logistic regression analysis was performed to investigate the potential determinants of the different stages of hypertension. Odds ratios (ORs) and 95% confidence intervals (CIs) were calculated. Statistical significance was set at $p < 0.05$.

### Ethical approval

As King Khalid Military Hospital is a part of King Salman Armed Forces Hospital, a single research and ethics committee is responsible for all ethical concerns of both the hospitals. Ethical approval for this study was obtained from the local research and ethics committee at King Salman Armed Forces Hospital under project number KSAFH-REC-2020-375. The informed consent was waived due to the retrospective nature of the study and the analysis used anonymous clinical data.

## Results

Data from the last three separate medical visits of all known hypertensive patients between 1 January 2019 and 31 December 2019 were collected. The variables utilized in the study were defined by the author in the data collection tool. The flow chart below shows the collected data (Fig 1).

Table 1 shows the overall profile of the 884 patients included in the final analysis. Patients were grouped according to stages of hypertension. More than half of the patients were aged between 38–47 years (52.0%). The majority of the patients were males (66.5%). More than 90% of the patients were Saudi Arabian (93.4%). Almost two-fifths of the patients had a positive family history of hypertension (44.6%). Four out of ten patients did not engage in any form of physical activity (42.0%).

Table 2 shows the mean systolic and diastolic blood pressures (mmHg) at different visits among the studied population.

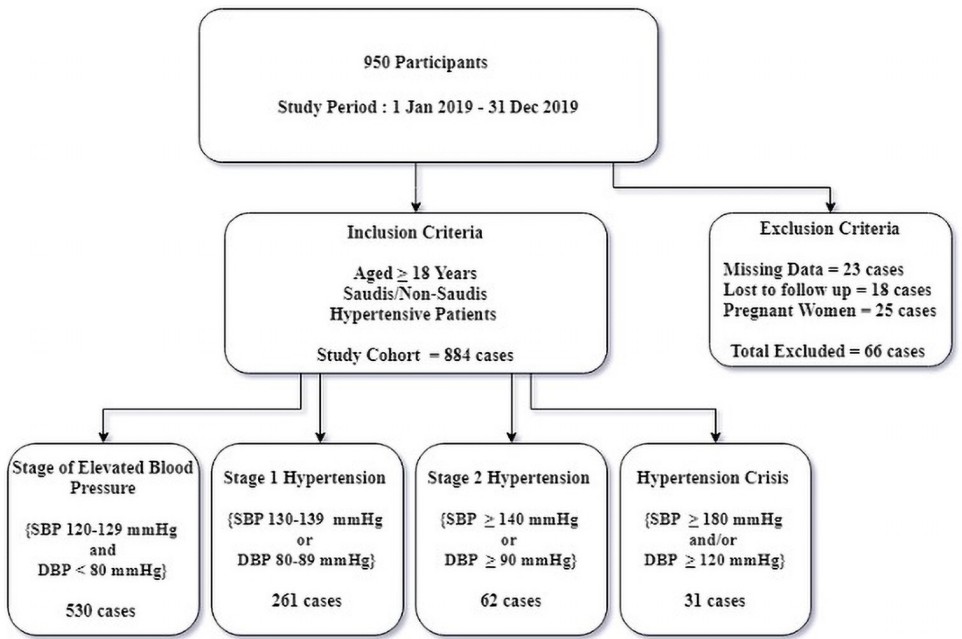

**Fig 1. Flow chart of the study cohort.**

**Table 1. Selected characteristics of the study population grouped by stages of hypertension (n = 884).**

| Variables | Stage of Elevated BP 530 (%) | Stage 1 HTN 261 (%) | Stage 2 HTN 62 (%) | HTN Crisis 31 (%) | Total 884 (%) |
|---|---|---|---|---|---|
| **Age** | | | | | |
| 18–27 years | 38 (7.2) | 8 (3.1) | 4 (6.5) | 3 (9.7) | 53 (6.0) |
| 28–37 years | 84 (15.8) | 11 (4.2) | 10 (16.1) | 5 (16.1) | 110 (12.4) |
| 38–47 years | 344 (64.9) | 56 (21.5) | 44 (71.0) | 16 (51.6) | 460 (52.0) |
| ≥ 48 years | 64 (12.1) | 186 (71.3) | 4 (6.5) | 7 (22.6) | 261(29.5) |
| **Sex** | | | | | |
| Male | 337 (63.6) | 195 (74.7) | 39 (62.9) | 17 (54.8) | 588 (66.5) |
| Female | 193 (36.4) | 66 (25.3) | 23 (37.1) | 14 (45.2) | 296 (33.5) |
| **Marital status** | | | | | |
| Single | 72 (13.6) | 30 (11.5) | 15 (24.2) | 9 (29.0) | 126 (14.3) |
| Married | 423 (79.8) | 197 (75.5) | 34 (54.8) | 14 (45.2) | 688 (75.6) |
| Divorced | 27 (5.1) | 23 (8.8) | 10 (16.1) | 3 (9.7) | 63 (7.1) |
| Widowed | 8 (1.5) | 11 (4.2) | 3 (4.8) | 5 (16.1) | 27 (3.1) |
| **Nationality** | | | | | |
| Saudi | 493 (93.0) | 248 (95.0) | 57 (91.9) | 28 (90.3) | 826 (93.4) |
| Non-Saudi | 37 (7.0) | 13 (5.0) | 5 (8.1) | 3 (9.7) | 58 (6.6) |
| **Employment** | | | | | |
| Employed | 386 (72.8) | 193 (73.9) | 51 (82.3) | 19 (61.3) | 649 (73.4) |
| Unemployed | 144 (27.2) | 68 (26.1) | 11 (17.7) | 12 (38.7) | 235 (26.6) |
| **Family History of HTN** | | | | | |
| Yes | 192 (36.2) | 133 (51.0) | 44 (71.0) | 25 (80.6) | 394 (44.6) |
| No | 338 (63.8) | 128 (49.0) | 18 (28.0) | 6 (19.4) | 490 (55.4) |
| **Smoking status** | | | | | |
| Never Smoked | 214 (40.4) | 83 (31.8) | 31 (50.0) | 16 (51.6) | 344 (38.9) |
| Past Smoker | 121 (22.8) | 69 (26.4) | 16 (25.8) | 7 (22.6) | 213 (24.1) |
| Current Smoker | 195 (36.8) | 109 (41.8) | 15 (24.2) | 8 (25.8) | 327 (37.0) |
| **Physical activity** | | | | | |
| Yes | 431 (81.3) | 59 (22.6) | 15 (24.2) | 8 (25.8) | 513 (58.0) |
| No | 99 (18.7) | 202 (77.4) | 47 (75.8) | 23 (74.2) | 371 (42.0) |

Abbreviations: BP-blood pressure, HTN-Hypertension.

Fig 2 shows the incidence of hypertension within the four stages of BP defined in this study. Stage of elevated BP: SBP 120–129 mmHg and DBP<80 mmHg; stage1: SBP 130–139 mmHg or DBP 80–89 mmHg; stage 2: SBP≥140 mmHg or DBP≥90 mmHg; hypertension crisis: SBP > 180 mmHg and/or DBP > 120 mmHg. The incidence rate was reported as per 1,000

**Table 2. Mean systolic and diastolic blood pressure at different visits (mm hg) and incidence of hypertension.**

| Stages of Hypertension | Visit 1 | | Visit 2 | | Visit 3 | |
|---|---|---|---|---|---|---|
| | SBP (Mean ± SD) | DBP (Mean ± SD) | SBP (Mean ± SD) | DBP (Mean ± SD) | SBP (Mean ± SD) | DBP (Mean ± SD) |
| Stage of Elevated BP | 122.66 ± 5.23 | 74.22 ± 2.81 | 123.83 ± 5.44 | 73.33 ± 2.98 | 123.91 ± 5.56 | 74.60 ± 2.89 |
| Stage 1 HTN | 132.05 ± 5.42 | 84.53 ± 2.78 | 134.85 ± 5.79 | 85.36 ± 2.88 | 133.98 ± 5.83 | 83.68 ± 2.92 |
| Stage 2 HTN | 146.01 ± 3.29 | 92.52 ± 3.22 | 150.70 ± 3.06 | 93.02 ± 3.28 | 145.78 ± 3.01 | 98.33 ± 2.84 |
| Hypertension crisis | 180.26 ± 4.36 | 125.78 ± 3.76 | 189.64 ± 5.63 | 132.16 ± 4.97 | 185.37 ± 5.19 | 127.96 ± 3.48 |

Abbreviations: BP-blood pressure, SBP-systolic blood pressure, DBP- diastolic blood pressure, HT-hypertension, SD- standard deviation, CI-confidence interval.

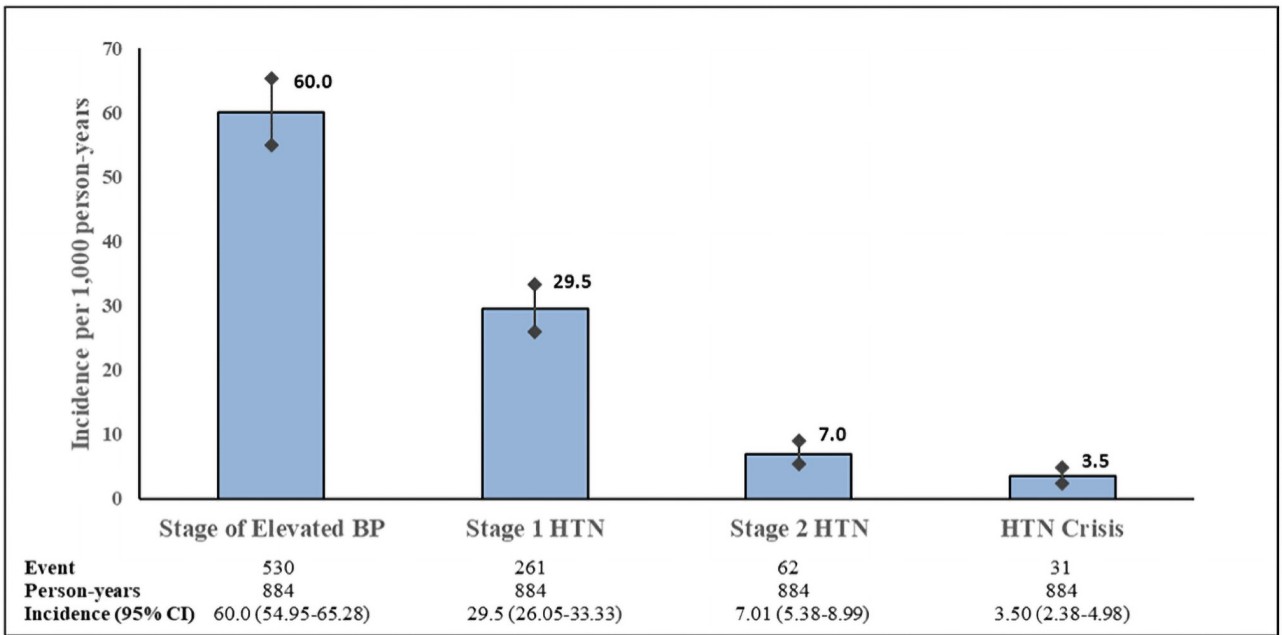

**Fig 2. Incidence of different stages of hypertension and corresponding 95% confidence intervals.**

person-years. Error bars represent 95% confidence intervals for the incidence rate (Fig 2). The incidence rates according to BP stage were as follows:

Incidence of stage of elevated BP = 530/884 = 60.0 new cases of stage of elevated BP per 1000 persons.

Incidence of stage 1 hypertension = 261/884 = 29.5 new cases of stage 1 hypertension per 1000 persons.

Incidence of stage 2 hypertension = 62/884 = 7.01 new cases of stage 2 hypertension per 1000 persons.

Incidence of hypertension crisis = 31/884 = 3.50 new cases of hypertension crisis per 1000 persons.

Table 3 shows the relative risk of different stages of hypertension by age, sex, and family history of hypertension. The risk of hypertension crisis was found to increase by 5.9 times in old age ($\geq$48 years) compared to young age. Males were 1.6 times more likely to develop stage 2 hypertension than females. A positive family history of hypertension was associated with a 151% increase in the risk of hypertension crisis (Table 3).

Table 4 reports the output of the multinomial logistic regression model. The main effect model was used on different stages of hypertension by considering the above stated risk factors. The results of the analysis indicated that the fitted model was good, with $\chi^2$ = 594.97, p-value = 0.992 (deviance criterion used), and Nagelkerke $R^2$ = 0.731. Additionally, the test of the full model against the constant model showed a significant result, $\chi^2$ = 772.87, p-value < 0.0001, suggesting that the above mentioned risk factors as a group satisfactorily distinguished the four stages of hypertension.

**Table 3. Relative risk of hypertension by age, sex, and family history of hypertension.**

| Risk Factors | Relative Risk (95% CI for RR) | | |
|---|---|---|---|
| | Stage 1 HTN | Stage 2 HTN | Hypertension Crisis |
| **Age** | | | |
| 18–27 years | 1.21 (1.01–1.45) | 0.49 (0.26–0.95) | 1.28 (0.62–2.62) |
| 28–37 years | 1.33 (1.17–1.49) | 0.31 (0.17–0.54) | 1.35 (0.81–2.27) |
| 38–47 years | 0.97 (0.87–1.06) | 0.25 (0.19–0.32) | 1.68 (1.11–2.51) |
| ≥48 years | 0.33 (0.26–0.41) | 0.32 (0.17–0.59) | 5.92 (4.72–7.42) |
| **Sex** | | | |
| Male | 0.82 (0.68–0.98) | 1.67 (0.53–2.85) | 1.37 (0.93–2.02) |
| Female | 1.13 (1.01–1.26) | 1.17 (1.07–1.27) | 0.96 (0.92–1.01) |
| **Family History of Hypertension** | | | |
| Yes | 0.77 (0.69–0.87) | 1.41 (1.12–1.74) | 1.51 (1.0.3–2.23) |
| No | 1.44 (1.23–1.69) | 0.86 (0.79–0.94) | 0.95 (0.91–0.99) |

Abbreviations: RR-relative risk, CI-confidence interval, HTN-hypertension. Note: The reference category is the stage of elevated blood pressure.

**Table 4. Multinomial logistic regression differentiating stage of elevated blood pressure from stage 1 hypertension, stage 2 hypertension, and hypertension crisis.**

| Variables | Categories | Stage 1 Hypertension | | | | Stage 2 Hypertension | | | | Hypertension Crisis | | | |
|---|---|---|---|---|---|---|---|---|---|---|---|---|---|
| | | B | Wald $\chi^2$-Statistics | OR (95% CI) | P-value | B | Wald $\chi^2$-Statistics | OR (95% CI) | P-value | B | Wald $\chi^2$-Statistics | OR (95% CI) | P-value |
| **Age** | 28–37 years | 0.71 | 2.48 | 0.65 (0.23–1.75) | 0.369 | 0.49 | 1.96 | 0.78 (0.34–1.75) | 0.375 | 0.53 | 1.74 | 0.89 (0.33–2.42) | 0.271 |
| | 38–47 years | 0.99 | 3.49 | 0.63 (0.26–1.49) | 0.496 | 1.78 | 8.90 | 1.93 (0.26–2.49) | 0.293 | 1.79 | 2.82 | 1.58 (0.41–2.42)* | 0.026* |
| | ≥ 48 years | 1.29 | 3.91 | 5.00 (1.63–15.36)* | 0.01* | 1.93 | 4.36 | 6.00 (2.48–9.56)* | 0.005* | 2.17 | 8.68 | 3.62 (1.99–8.42)* | 0.04* |
| **Sex** | Male | 1.71 | 10.44 | 5.54 (1.96–15.67)* | 0.001* | 1.67 | 7.36 | 3.69 (1.68–7.78)* | <0.001* | 1.21 | 4.61 | 2.84 (0.57–5.92)* | 0.001* |
| **Marital status** | Married | 1.56 | 10.23 | 1.20 (0.67–2.15)* | 0.478 | 1.31 | 6.13 | 1.58 (0.89–3.25)* | 0.001* | 1.79 | 5.21 | 1.83 (0.43–3.64)* | <0.001* |
| | Divorced | 1.63 | 5.81 | 1.02 (1.15–7.04)* | 0.013* | 1.54 | 7.32 | 1.62 (1.46–6.42)* | 0.02* | 1.58 | 2.73 | 0.55 (0.14–2.06) | 0.789 |
| | Widowed | 0.76 | 1.83 | 2.16 (0.55–8.47) | 0.371 | 0.94 | 1.36 | 3.13 (0.63–7.93) | 0.74 | 1.48 | 5.39 | 1.24 (0.26–5.90)* | <0.001* |
| **Nationality** | Saudi | 0.57 | 0.62 | 3.24 (0.89–11.73) | 0.401 | 0.71 | 0.89 | 6.24 (1.37–12.39) | 0.879 | 1.11 | 1.49 | 0.31 (0.09–1.08) | 0.065 |
| **Employment** | Unemployed | 0.31 | 0.74 | 0.49 (0.23–1.05) | 0.318 | 0.48 | 0.91 | 0.91 (0.41–1.29) | 0.53 | 0.53 | 1.73 | 0.69 (0.30–1.58) | 0.380 |
| **Family History of Hypertension** | Yes | 0.85 | 18.19 | 2.35 (1.59–5.48)* | 0.008* | 0.70 | 12.34 | 3.78 (1.23–6.74)* | <0.001* | 0.67 | 8.09 | 1.95 (1.23–3.09)* | 0.004* |
| **Smoking status** | Past Smoker | -0.67 | 1.99 | 0.51 (0.20–1.30) | 0.254 | -0.56 | 2.31 | 0.70 (0.16–2.96) | 0.158 | -0.74 | 1.89 | 0.47 (0.16–1.37) | 0.169 |
| | Current Smoker | 1.35 | 0.60 | 1.70 (0.29–3.72)* | *0.036 | 2.64 | 0.37 | 2.38 (0.87–6.21)* | 0.023* | 3.44 | 7.74 | 1.74 (1.23–4.76)* | 0.002* |
| **Physical activity** | No | 1.12 | 9.56 | 3.01 (1.51–6.27)* | *0.02 | 1.82 | 8.23 | 2.51 (1.12–5.97)* | 0.002* | 1.64 | 10.28 | 6.48 (2.46–9.14)* | 0.035* |

Reference categories: 1) Hypertension: stage of elevated blood pressure; 2) Age: 18–27 years, 3) Sex: female; 4) Marital status: single; 5) Nationality: non-Saudi; 6) Employment: employed; 7) Family history of hypertension: no; 8) Smoking status: never smoked; 9) Physical activity: yes.

Abbreviations: OR-Odds Ratio, CI-Confidence Interval.

Note:

* represents significant p-value <0.05.

The analysis revealed that the increasing risk of hypertension with age was apparent only when comparing participants in the upper age stratum ($\geq$48 years) with those in the lower stratum ($<$28 years). Male subjects were more likely to have stage 1 (OR = 5.54, 95% CI: 1.96–15.67) and stage 2 (OR = 3.69, 95% CI: 1.68–7.78) hypertension. Subjects with a positive family history of hypertension were more prone to develop stage 1 hypertension (OR = 2.35, 95% CI: 1.59–5.48). Lastly, the subjects who had no physical activity were more likely to develop stage 1 hypertension (OR = 3.01, 95% CI: 1.51–6.27) and hypertension crisis (OR = 6.48, 95% CI: 2.46–9.14) (Table 4).

## Discussion

This retrospective cohort study is among the few to determine the incidence of different stages of hypertension in a broadly inclusive population of subjects recruited into medical care at military-setting hospitals. The major findings of this study included the following: (1) the incidence of stage of elevated BP (60.0 cases per 1000 persons) during one year of the study period and (2) older age, male sex, marital status, positive family history, current smoking status, and physical inactivity were significantly associated with an increased risk of hypertension in the study population.

The present study revealed that the incidence of hypertension was high among the studied populations. This finding agrees with results from previous studies conducted in the Gulf region and other developing countries [18, 19]. An increase in BP with age is considered a universal aspect of human aging and the present study supported the fact that advancing age is an important risk factor for hypertension. This finding was also in agreement with results from previous studies conducted in developed and developing countries with different sociodemographics, varied cultures, and geographically diverse populations [20–22].

Globally, the impact of sex on the incidence of hypertension has been reported in several studies [23, 24]. This study explored the association between sex and hypertension. Men were more likely to develop hypertension than women. This result confirmed the findings of other regional and national studies that described the tendency of males to have a higher BP [25, 26]. The present study did not show any association between nationality and hypertension. According to this study, the incidence of hypertension was not linked to employment status. This was contrary to what has been reported in the previous studies regarding this association [27].

In addition, our results revealed the association between the incidence of hypertension and marital status. Married individuals were more likely to have stage 2 hypertension and a hypertension crisis. Previous studies have shown that hypertension is significantly more common among married and divorced individuals [28]. In contrast, some studies have found that never-married individuals are at a high risk of developing hypertension [29, 30]. Moreover, the present study highlighted that genetic factors are important risk factors for hypertension. Individuals with a positive family history of hypertension were discovered to be more likely to develop stage 2 hypertension crisis in the current study. This result is in line with other studies that reported that the involvement of genetic factors may cause an individual to be at increased risk of developing hypertension [31, 32].

Smoking status was another risk factor associated with the incidence of hypertension in this study. Population-based research studies have found that the incidence of hypertension is higher among current smokers [33, 34]. Lack of physical activity was also identified as an important modifiable risk factor and it was revealed that individuals with a sedentary lifestyle were more likely to develop hypertension. This finding supports the results of previous studies showing that physical inactivity increases the risk of hypertension [35, 36]. The shift in

hypertension from one stage to another could be attributed to poor adherence to medication or any other poor lifestyle habit.

### Strengths and limitations of the study

Most previous studies have investigated predictors of hypertension using separate models. However, this study investigated four stages of hypertension using the same model. The major strength of this study is that it is a cohort study which attempted to establish how the associations between the outcomes (stages of hypertension) and the suspected predictors vary according to the level of outcome. This study also has its limitations. Not all risk factors related to hypertension were included in this study due to resource constraints. In addition, the current study did not include data related to hypertension, such as symptoms, check-up visits, elevated blood lipids, circulating inflammatory markers, oxidative stress markers, antihypertensive treatment, etc. There could have been a misclassification bias created in abstracting records which was overcome by cross-checking the data entered by the two investigators.

## Conclusion

The present study demonstrated that the incidence of stage of elevated BP was high among the studied patients at military-setting hospitals in Tabuk, Saudi Arabia. Age, sex, marital status, family history, smoking status, and physical activity were significant predictors of hypertension development. The findings of this study will bridge the existing knowledge gap in literature regarding hypertension and will be of significance not only for the region of study but also for the other parts of Saudi Arabia, the Gulf region, and possibly other developing countries.

We proposed a logistic regression model to predict the progression of different stages of hypertension in this study. These results strongly support the adoption of standardised systemic hypertension screening for patients and highlight the need to introduce evidence-based noncommunicable disease prevention approaches in the Tabuk region of Saudi Arabia. Further prospective multicentre cohort studies evaluating hypertension and its complications in Saudi Arabia are highly recommended.

## Acknowledgments

We would like to acknowledge King Salam Armed Forces Hospital and King Khalid Military Hospital for providing the opportunity to conduct this study. We also extend our gratitude to Mr. Marwan Manajreh for his dedicated support during the data collection and entry process.

## Author Contributions

**Conceptualization:** Umar Yagoub, Nasrin S. Saiyed.

**Data curation:** Umar Yagoub, Nasrin S. Saiyed.

**Formal analysis:** Umar Yagoub, Nasrin S. Saiyed.

**Funding acquisition:** Nasrin S. Saiyed, Bandar Al Qahtani, Attiya Mohammed Al Zahrani.

**Investigation:** Umar Yagoub, Bandar Al Qahtani, Yassir Birema, Ibrahim Al Hariri.

**Methodology:** Umar Yagoub, Nasrin S. Saiyed.

**Project administration:** Umar Yagoub, Nasrin S. Saiyed, Bandar Al Qahtani, Attiya Mohammed Al Zahrani.

**Resources:** Attiya Mohammed Al Zahrani.

**Software:** Umar Yagoub, Nasrin S. Saiyed.

**Supervision:** Umar Yagoub, Nasrin S. Saiyed, Bandar Al Qahtani, Attiya Mohammed Al Zahrani, Yassir Birema, Ibrahim Al Hariri.

**Validation:** Umar Yagoub, Nasrin S. Saiyed.

**Visualization:** Umar Yagoub, Nasrin S. Saiyed.

**Writing – original draft:** Umar Yagoub, Nasrin S. Saiyed.

**Writing – review & editing:** Umar Yagoub, Nasrin S. Saiyed.

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
