## [Decision Letter · Decision Letter 0]

16 Apr 2021

PONE-D-21-07004

Incidence and Risk Factors of Hypertension: A Multicentre Retrospective Cohort Study in Tabuk, Saudi Arabia

PLOS ONE

Dear Dr. Mohamed,

Thank you for submitting your manuscript to PLOS ONE. After careful consideration, we feel that it has merit but does not fully meet PLOS ONE’s publication criteria as it currently stands. Therefore, we invite you to submit a revised version of the manuscript that addresses all the points raised during the review process.

We look forward to receiving your revised manuscript.

Kind regards,

Manal S. Fawzy, Ph.D., M.D.

Academic Editor

PLOS ONE

Journal Requirements:

"The study was fully funded by Research Unit, Academic Affairs Department, King Salman Armed Forces Hospital Tabuk, Saudi Arabia."

"NO"

6. Please ensure that you refer to Figure 1 in your text as, if accepted, production will need this reference to link the reader to the figure.

Reviewers' comments:

Reviewer's Responses to Questions

**Comments to the Author**

1. Is the manuscript technically sound, and do the data support the conclusions?

Reviewer #1: Yes

Reviewer #2: Partly

2. Has the statistical analysis been performed appropriately and rigorously? 

Reviewer #1: Yes

Reviewer #2: No

3. Have the authors made all data underlying the findings in their manuscript fully available?

Reviewer #1: Yes

Reviewer #2: No

4. Is the manuscript presented in an intelligible fashion and written in standard English?

Reviewer #1: Yes

Reviewer #2: No

5. Review Comments to the Author

Reviewer #1: Thank you for a thorough study of the epidemiology of hypertension in an understudied population. I have a few comments:

-First of all, the paper needs thorough language editing.

-In the methodology, it is stated that patients were included with hypertension diagnosed according to ICD-10. ICD-10 is a disease coding/classification system, not a set of diagnostic criteria. The criteria used (ACC/AHA 2017) should be mentioned instead.

-I couldn't find a clear mention of how long the (mean) follow up interval was anywhere. Please include, because a short interval would be a limitation that needs mentioning.

-References should be provided for the parameters used in the sample size calculation such as the odds ratio, risk/prevalence ratio, etc, or at least some justification for their choice if arbitrary.

-Since this is effectively a time-series analysis, I think a multivariate cox proportionate hazards model would be appropriate to include to help disentangle the effect of confounding variables.

-In the results, I couldn't help but notice that so few patients are over 47, which is not very representative of the hypertensive patient population. The reason for this should be clarified.

-Some essential data such as the presence and type of antihypertensive treatment are not included in the analysis. This should at least be mentioned in the limitations. It would also be useful to mention the reason for presentation, e.g. checkup visit, symptoms, etc if possible.

Overall, the study provides much needed data about hypertensive risk factors in the Saudi Arabian population, and would be useful to publish after the above considerations.

Reviewer #2: 1- GENERAL COMMENT

The strength of the current research is studying all the stages of hypertension aiming to identify the incidence rate and the predictors associated with hypertension, which is an interesting subject. There are very few studies dealing with the incidence of hypertension and changes in blood pressure over time. although the difficulty of studying the incidence rate of hypertension lie in the diversity of the incidence rate of hypertension with the regions.

The manuscript needs English editing since there are several typos and grammatical errors.

2- INTRODUCTION

2.1. The authors should elaborate more on the previous research that includes the incidence of hypertension in different ethnicities. For example, the CARLA-Cohort study.

Lacruz ME, Kluttig A, Hartwig S, Löer M, Tiller D, Greiser KH, Werdan K, Haerting J. Prevalence and Incidence of Hypertension in the General Adult Population: Results of the CARLA-Cohort Study. Medicine (Baltimore). 2015 Jun;94(22):e952. doi: 10.1097/MD.0000000000000952. PMID: 26039136

2.2. It is advised to cite previous studies about the incidence rate of hypertension in the Gulf area.

3. MATERIALS AND METHODS

3.1. The methods were not written in sufficient detail to allow the study to be replicated. The study design was reported as a retrospective cohort, therefore this should be explained, and the two cohorts should be defined; the exposed cohort and the non-exposed one.

3.1. The recruitment of the patients was described as from the nephrology, cardiology and internal medicine departments. Was the recruitment from the outpatient clinics or the inpatient department? This issue was not clear.

3.2. The patients had three blood pressure measurements, if the patient has more than 3 measurements, what did the authors do? It should be stated that the last three measurements were used in the methods section. Did you mean these measurements? What is the minimum interval allowed between these three measurements? All these details should be mentioned in the methods section.

3.3. Why did the authors exclude the diagnosis of hypertension in less than 6 months? This might bias the results.

3.4. Regarding the operational definitions, the referee suggests removing the paragraph that defines the blood pressure, which is found on page 2, line 41-44.

3.5. The inclusion criteria were not clear in the Methods section. It should be written in details.

3.6. How did the authors adjust for the change of age overtime of the study of one year?

3.7. Regarding the calculation of the incidence rate,

• The equations of incidence rate and relative risk should be supported by the related references.

• The way of calculation of the incidence rate should be written in the methods more comprehensively, and in the results as well. For example, how did the authors diagnose the new cases of hypertension? How many new cases did the authors diagnose? Then, put the incidence rate equation defining both the numerator and denominator, to find out the incidence rate.

3.8. All the risk factors should be defined and cited. For example, the physical activity was defined but not referenced, while the family history of hypertension, smoking status were not defined nor referenced. The referee suggests applying these in the operational definition.

4. RESULTS

4.1. The stage of elevated BP according to ACC guidelines 2017, is stated in different contexts of the manuscript as ELEVATED HYPERTENSION, which needs to be corrected through the manuscript to THE STAGE OF ELEVATED BP, because it does not fulfil the criteria of hypertension. The same error was found in figure 1.

4.2. In all stages of BP STATED in figure 1, the total cases were demonstrated. The new cases should be mentioned as well.

4.3. The legend of Table 2 stated the mean systolic and diastolic BP and the incidence of hypertension. There is no incidence of hypertension in this table.

4.4. In Figure 2, it is advised to highlight the equation of each stage defining the numerator and denominator till you calculate the incidence rate. Importantly, mention the new cases.

4.5. On page 7, it was mentioned that there is an increasing risk of stage 2 with age. Nevertheless, not only stage 2, but also stage 1 and the hypertensive crisis.

4.6. Table 4 needs important modifications:

4.6.1. It is very important to mention the p-value for each relation of the table, and highlighting the significant ones.

4.6.2. The table depicted 4 relationships as significant, meanwhile, they need to be double-checked for the significance because the CI of the odds ratio (OR) ranging from below 1 to above 1, which in many situations refers to a non-significant relationship. It is advised to consult a statistician to solve this issue. The four relations are the marital status and stage 1, the marital status and hypertensive crisis, the current smoking and stage one, and the current smoking and stage 2.

4.6.3. Why the elevated BP stage was not included in table 3 and 4. It is better to be added.

5. DISCUSSION

5.1. It needs to be comprehensive.

5.2. All the risk factors should be discussed in more details. As unemployment and nationality/race were not mentioned.

5.3. It will be valuable to mention the black race as an important subgroup.

5.4. All the references used in the discussion should be explained in detail trying to explain the similarities and differences with the current study.

6. CONCLUSION

6.1. It should be reconstructed to highlight the main findings and support the study results. The conclusion in the abstract should be matched with the manuscript conclusion.

6. PLOS authors have the option to publish the peer review history of their article (what does this mean?). If published, this will include your full peer review and any attached files.

Reviewer #1: No

Reviewer #2: **Yes: **Hussein M. Ismail

---

## [Author Response · Author response to Decision Letter 0]

14 Nov 2021

08/11/2021

Emily Chenette

Editor-in-Chief

PLOS One

Dear Editor:

I wish to resubmit a research article for publication in PLOS One, titled “Investigating the incidence and risk factors of hypertension: a multicentre retrospective cohort study in Tabuk, Saudi Arabia.” The paper was coauthored by Nasrin S Saiyed, Bandar Al Qahtani, Attiya Mohammed Al Zahrani, Yassir Birema, and Ibrahim Al Hariri. 

This study aimed to determine the incidence of different stages of hypertension and its associated modifiable and non-modifiable risk factors among patients attending two different hospitals in Tabuk, Saudi Arabia. We believe that our study makes a significant contribution to the literature because there have been studies in the past in Saudi Arabian populations exploring the relevant risk factors for hypertension, but all these studies were cross-sectional in design. To the best of our knowledge, the current retrospective cohort study on hypertension and related risk factors is the first study of this type to be conducted in the region of Tabuk in Saudi Arabia. 

Further, we believe that this paper will be of interest to the readership of your journal because the results of this study can be extended and applied to other regions of Saudi Arabia as well as in the Gulf region which has similar demographics and an increased incidence of diseases which are complicated by hypertension. In addition, we have also proposed a logistic regression model which can predict the progression of different stages of hypertension and will be highly beneficial for medical practitioners.

We appreciate the time and effort that you and the reviewers have dedicated to providing your valuable feedback on my manuscript. We are grateful to the reviewers for their insightful comments on my paper. We have been able to incorporate changes to reflect most of the suggestions provided by the reviewers. We have highlighted the changes within the manuscript. Here is a point-by-point response to the reviewers’ comments and concerns.

Comment 1: Thank you for stating the following in the Funding Section of your manuscript: "The study was fully funded by Research Unit, Academic Affairs Department, King Salman Armed Forces Hospital Tabuk, Saudi Arabia." We note that you have provided funding information that is not currently declared in your Funding Statement. However, funding information should not appear in the Acknowledgments section or other areas of your manuscript. We will only publish funding information present in the Funding Statement section of the online submission form. Please remove any funding-related text from the manuscript and let us know how you would like to update your Funding Statement. Currently, your Funding Statement reads as follows: "NO" Please include your amended statements within your cover letter; we will change the online submission form on your behalf.

Response: The statement related to funding was removed from the manuscript. The Funding Statement for the online submission is as follows: 

“The study was fully funded by Research Unit, Academic Affairs Department, King Salman Armed Forces Hospital Tabuk, Kingdom of Saudi Arabia.”

Comment 2: We note that you have indicated that data from this study are available upon request. PLOS only allows data to be available upon request if there are legal or ethical restrictions on sharing data publicly. For information on unacceptable data access restrictions, please see http://journals.plos.org/plosone/s/dataavailability#loc-unacceptable-data-access-restrictions.

Response: The data used to support the findings of this study are available from King Salman Armed Forces Hospital and King Khalid Military Hospital but restrictions apply to the availability of these data as it contains potentially identifying patient information. Data are however available from first author upon reasonable request and with permission of research and ethics committee at King Salman Armed Forces Hospital. The data sharing statement is as follows:

“The data used to support the findings of this study are available from King Salman Armed Forces Hospital and King Khalid Military Hospital but restrictions apply to the availability of these data, which were used under license for the current study, and so are not publicly available. Data are however available from first author upon reasonable request and with permission of research and ethics committee at King Salman Armed Forces Hospital.”

Comment 3. PLOS requires an ORCID iD for the corresponding author in Editorial Manager on papers submitted after December 6th, 2016. Please ensure that you have an ORCID iD and that it is validated in Editorial Manager. To do this, go to ‘Update my Information’ (in the upper left-hand corner of the main menu), and click on the Fetch/Validate link next to the ORCID field. This will take you to the ORCID site and allow you to create a new iD or authenticate a pre-existing iD in Editorial Manager. Please see the following video for instructions on linking an ORCID iD to your Editorial Manager account: https://www.youtube.com/watch? v=_xcclfuvtxQ 

Response: The ORCID id is updated. The ORCID id for the corresponding author Dr. Umar Yagoub is 0000-0003-1427-251X.

Comment 4. Your ethics statement should only appear in the Methods section of your manuscript. If your ethics statement is written in any section besides the Methods, please move it to the Methods section and delete it from any other section. Please ensure that your ethics statement is included in your manuscript, as the ethics statement entered into the online submission form will not be published alongside your manuscript.

Response: The ethics statement is moved to the Methods section of the manuscript.

Comment 5. Please ensure that you refer to Figure 1 in your text as, if accepted, production will need this reference to link the reader to the figure.

Response: Change has been made to the manuscript. The figure is referred in the text. 

Comment 6. Is the manuscript presented in an intelligible fashion and written in standard English? / the Paper needs thorough language editing.

Response: The manuscript has been edited by professional language editor for English editing. The language editing certificate is attached with this email.

Comment 7. In the methodology, it is stated that patients were included with hypertension diagnosed according to ICD10. ICD-10 is a disease coding/classification system, not a set of diagnostic criteria. The criteria used (ACC/AHA 2017) should be mentioned instead.

Response: Correction is made in the manuscript. All patients aged 18 years and older with diagnosed hypertension according to American College of Cardiology/American Heart Association (ACC/AHA 2017) were included in the study cohort. Whose disease coding were done according to ICD‐10 (International Statistical Classification of Diseases and Related Health Problems 10th Revision). 

Comment 8. I couldn't find a clear mention of how long the (mean) follow up interval was anywhere. Please include, because a short interval would be a limitation that needs mentioning.

Response: The mean follow-up interval for the current study was 3 months. 

Comment 9. References should be provided for the parameters used in the sample size calculation such as the odds ratio, risk/prevalence ratio, etc., or at least some justification for their choice if arbitrary. 

Response: Limited resources were the primary reason for the choice of the sample size. 

Comment 10. Since this is effectively a time-series analysis, I think a multivariate cox proportionate hazards model would be appropriate to include to help disentangle the effect of confounding variables. 

Response: Here the logistic regression analysis is used because the follow-up for all the patients is complete and have no censored data. Also, the logistic regression result can be presented to better visualize the differences in the number of events between groups.

Comment 11. In the results, I couldn't help but notice that so few patients are over 47, which is not very representative of the hypertensive patient population. The reason for this should be clarified. 

Response: Overall, 81.5% patients in the cohort aged 38 years and older had hypertension in the current study, which shows a representative of the hypertensive patient population. 

Comment 12. Some essential data such as the presence and type of antihypertensive treatment are not included in the analysis. This should at least be mentioned in the limitations. It would also be useful to mention the reason for presentation, e.g. checkup visit, symptoms, etc. if possible. 

Response: As due to resources constraints we didn’t collect the data related checkup visit, symptoms, the presence and type of antihypertensive treatment for the current study. These are now added in the limitations. 

Comment 13. The authors should elaborate more on the previous research that includes the incidence of hypertension in different ethnicities. For example, the CARLA-Cohort study.

Response: Comparisons between different studies reporting incidence rates of hypertension are problematic due to diverse definitions used and differences in the population structure, especially, variances in age range. The CARLA-Cohort study is discussed in the introduction. 

Comment 14. It is advised to cite previous studies about the incidence rate of hypertension in the Gulf area.

Response: Cited in the Discussion Section. 

Comment 15. The methods were not written in sufficient detail to allow the study to be replicated. The study design was reported as a retrospective cohort, therefore this should be explained, and the two cohorts should be defined; the exposed cohort and the non-exposed one.

Response: Changes have been made in the manuscript. 

Comment 16. The recruitment of the patients was described as from the nephrology, cardiology and internal medicine departments. Was the recruitment from the outpatient clinics or the inpatient department? This issue was not clear.

Response: The patients were recruited from both outpatient clinics and the inpatient departments. Added in the methodology. 

Comment 17. The patients had three blood pressure measurements, if the patient has more than 3 measurements, what did the authors do? It should be stated that the last three measurements were used in the methods section. Did you mean these measurements? What is the minimum interval allowed between these three measurements? All these details should be mentioned in the methods section. 

Response: The patients had three blood pressure measurements, if the patient has more than 3 measurements then only last three measurements were used for the study. The rest blood pressure measurements did not consider in the study. The changes in the methodology is made. 

Comment 18. Why did the authors exclude the diagnosis of hypertension in less than 6 months? This might bias the results. 

Response: Most of the newly diagnosed hypertensive patients didn’t have three visits in 6 months that is the reason for exclusion. 

Comment 19. Regarding the operational definitions, the referee suggests removing the paragraph that defines the blood pressure, which is found on page 2, line 41-44. 

Response: Removed from the manuscript. 

Comment 20. The inclusion criteria were not clear in the Methods section. It should be written in details. 

Response: The inclusion criteria in the manuscript is rewritten. 

Comment 21. How did the authors adjust for the change of age over time of the study of one year? 

Response: Crude incidence rates are calculated for the same.

Comment 22. Regarding the calculation of the incidence rate, the equations of incidence rate and relative risk should be supported by the related references.

Response: References cited in the manuscript. 

Comment 23. The way of calculation of the incidence rate should be written in the methods more comprehensively, and in the results as well. For example, how did the authors diagnose the new cases of hypertension? How many new cases did the authors diagnose? Then, put the incidence rate equation defining both the numerator and denominator, to find out the incidence rate.

Response: When examining patients, all physicians have access to the hospital's electronic records. The electronic medical record in the hospital keeps track of a patient's previous visits, current medical conditions including their blood pressure readings, as well as their current drug list and comorbidities. Appropriate diagnosis of hypertension was defined as an ICD-10 code on the hospital electronic record system. 

Calculations for the incidence rates for the different stages of hypertension were made under the table no. 2. 

Comment 23. All the risk factors should be defined and cited. For example, the physical activity was defined but not referenced, while the family history of hypertension, smoking status were not defined nor referenced. The referee suggests applying these in the operational definition.

Response: Changes have been made in the manuscript. 

Comment 24. In the results section, the stage of elevated BP according to ACC guidelines 2017, is stated in different contexts of the manuscript as ELEVATED HYPERTENSION, which needs to be corrected through the manuscript to THE STAGE OF ELEVATED BP, because it does not fulfil the criteria of hypertension. The same error was found in figure 1.

Response: Agreed, the correction is made in the manuscript. 

Comment 25. In the results section, in all stages of BP STATED in figure 1, the total cases were demonstrated. The new cases should be mentioned as well. 

Response: In the figure 1, new cases were mentioned as events.

Comment 26. In the results section, the legend of Table 2 stated the mean systolic and diastolic BP and the incidence of hypertension. There is no incidence of hypertension in this table. In Figure 2, it is advised to highlight the equation of each stage defining the numerator and denominator till you calculate the incidence rate. Importantly, mention the new cases. 

Response: Agreed, the calculation of incidence rate is mentioned under table 2 with new cases for each stages of hypertension. 

Comment 27. On page 7, it was mentioned that there is an increasing risk of stage 2 with age. Nevertheless, not only stage 2, but also stage 1 and the hypertensive crisis. 

Response: Agreed, correction has been done in the manuscript. 

Comment 28. Table 4 needs important modifications: It is very important to mention the p-value for each relation of the table, and highlighting the significant ones. 

Response: The p-values for each categories is added in table. 

Comment 29. The table depicted 4 relationships as significant, meanwhile, they need to be double-checked for the significance because the CI of the odds ratio (OR) ranging from below 1 to above 1, which in many situations refers to a non-significant relationship. It is advised to consult a statistician to solve this issue. The four relations are the marital status and stage 1, the marital status and hypertensive crisis, the current smoking and stage one, and the current smoking and stage 2.

 Response: The justification from the statistician is quoted as follows: 

Two measures of 'significance' p value and the confidence interval. Although it may have met the threshold for p value 'statistical significance' what a small OR and CI touching or crossing 0 are concluding that the effect itself is very small.

Comment 30. Why the elevated BP stage was not included in table 3 and 4. It is better to be added. 

Response: The elevated BP stage was not included in table 3 and 4 because it is the reference category.

Comment 31. In the discussion section, it needs to be comprehensive. 

Response: Changes have been made in the discussion section of the manuscript. 

Comment 32. All the risk factors should be discussed in more details. As unemployment and nationality/race were not mentioned. 

Response: Changes have been made in the discussion section of the manuscript. 

Comment 33. It will be valuable to mention the black race as an important subgroup. 

Response: As we did not include the race category in the study we are unable to quote about the black race. Only nationality of the subject included in the research. Which is either Saudi or non-Saudi. 

Comment 34. All the references used in the discussion should be explained in detail trying to explain the similarities and differences with the current study. 

Response: Changes have been made in the discussion section of the manuscript. 

Comment 35. In the conclusion section, it should be reconstructed to highlight the main findings and support the study results. The conclusion in the abstract should be matched with the manuscript conclusion.

Response: Changes have been made in the discussion section of the manuscript. 

This manuscript has not been published or presented elsewhere in part or in entirety and is not under consideration by another journal. All study participants provided informed consent, and the study design was approved by the appropriate ethics review board. We have read and understood your journal’s policies, and we believe that neither the manuscript nor the study violates any of these. There are no conflicts of interest to declare.

Thank you for your consideration. I look forward to hearing from you.

Sincerely,

Umar Yagoub

Research Department, King Salman Armed Forces Hospital,

Tabuk, Saudi Arabia 

mohammedumar2001@yahoo.com

---

## [Decision Letter · Decision Letter 1]

21 Dec 2021

Investigating the incidence and risk factors of hypertension: a multicentre retrospective cohort study in Tabuk, Saudi Arabia

PONE-D-21-07004R1

Dear Dr. Mohamed,

We’re pleased to inform you that your manuscript has been judged scientifically suitable for publication and will be formally accepted for publication once it meets all outstanding technical requirements.

Kind regards,

Manal S. Fawzy, Ph.D., M.D.

Academic Editor

PLOS ONE

Additional Editor Comments (optional):

The authors have adequately addressed the concerns raised by the reviewers. Thank you

Reviewers' comments:

Reviewer's Responses to Questions

**Comments to the Author**

1. If the authors have adequately addressed your comments raised in a previous round of review and you feel that this manuscript is now acceptable for publication, you may indicate that here to bypass the “Comments to the Author” section, enter your conflict of interest statement in the “Confidential to Editor” section, and submit your "Accept" recommendation.

Reviewer #1: All comments have been addressed

Reviewer #2: All comments have been addressed

2. Is the manuscript technically sound, and do the data support the conclusions?

Reviewer #1: Yes

Reviewer #2: Yes

3. Has the statistical analysis been performed appropriately and rigorously? 

Reviewer #1: Yes

Reviewer #2: Yes

4. Have the authors made all data underlying the findings in their manuscript fully available?

Reviewer #1: Yes

Reviewer #2: Yes

5. Is the manuscript presented in an intelligible fashion and written in standard English?

Reviewer #1: Yes

Reviewer #2: Yes

6. Review Comments to the Author

Reviewer #1: (No Response)

Reviewer #2: (No Response)

7. PLOS authors have the option to publish the peer review history of their article (what does this mean?). If published, this will include your full peer review and any attached files.

Reviewer #1: No

Reviewer #2: **Yes: **Hussein M. Ismail

---

## [Editor Report · Acceptance letter]

27 Dec 2021

PONE-D-21-07004R1 

Investigating the incidence and risk factors of hypertension: a multicentre retrospective cohort study in Tabuk, Saudi Arabia 

Dear Dr. Yagoub:

I'm pleased to inform you that your manuscript has been deemed suitable for publication in PLOS ONE. Congratulations! Your manuscript is now with our production department. 

Kind regards, 

on behalf of

Professor Manal S. Fawzy 

Academic Editor

PLOS ONE